

# A secure solution based on load-balancing algorithms between regions in the cloud environment

Sarah Eljack[1], Mahdi Jemmali[1,2,3], Mohsen Denden[4,5], Sadok Turki[6], Wael M. Khedr[1,7], Abdullah M. Algashami[1] and Mutasim ALsadig[1]

[1] Department of Computer Science and Information, College of Science at Zulfi, Majmaah University, Majmaah, Saudi Arabia
[2] Mars Laboratory, University of Sousse, Sousse, Tunisia
[3] Department of Computer Science, Higher Institute of Computer Science and Mathematics, University of Monastir, Monastir, Tunisia
[4] Department of Computer and Information Technologies, College of Telecommunication and Information, Technical and Vocational Training Corporation TV TC, Riyadh CTI, Saudi Arabia
[5] Department of Computer Science, Higher Institute of Applied Sciences of Sousse, Sousse University, Sousse, Tunisia
[6] Department of Logistic and Maintenance, UFR MIM at Metz, University of Lorraine, Metz, France
[7] Department of Mathematics, Faculty of Science, Zagazig University, Zagazig, Egypt

Corresponding author
Sarah Eljack, s.alshiekh@mu.edu.sa

## ABSTRACT

The problem treated in this article is the storage of sensitive data in the cloud environment and how to choose regions and zones to minimize the number of transfer file events. Handling sensitive data in the global internet network many times can increase risks and minimize security levels. Our work consists of scheduling several files on the different regions based on the security and load balancing parameters in the cloud. Each file is characterized by its size. If data is misplaced from the start it will require a transfer from one region to another and sometimes from one area to another. The objective is to find a schedule that assigns these files to the appropriate region ensuring the load balancing executed in each region to guarantee the minimum number of migrations. This problem is NP-hard. A novel model regarding the regional security and load balancing of files in the cloud environment is proposed in this article. This model is based on the component called "Scheduler" which utilizes the proposed algorithms to solve the problem. This model is a secure solution to guarantee an efficient dispersion of the stored files to avoid the most storage in one region. Consequently, damage to this region does not cause a loss of big data. In addition, a novel method called the "Grouping method" is proposed. Several variants of the application of this method are utilized to propose novel algorithms for solving the studied problem. Initially, seven algorithms are proposed in this article. The experimental results show that there is no dominance between these algorithms. Therefore, three combinations of these seven algorithms generate three other algorithms with better results. Based on the dominance rule, only six algorithms are selected to discuss the performance of the proposed algorithms. Four classes of instances are generated to measure and test the performance of algorithms. In total, 1,360 instances are tested. Three metrics are used to assess the algorithms and make a comparison between them. The experimental results show that the best algorithm is the "Best-value of four algorithms" in 86.5% of cases with an average gap of 0.021 and an average running time of 0.0018 s.

# INTRODUCTION

Choosing regions or zones in the cloud environment aims to provide better performance and latency. Cost is also an important factor in choosing regions and zones in the cloud. Sometimes a business chooses the region based on the location of its customers. Load balancing is also an essential factor as it improves data security. In fact, the sensitivity of applications hosted in the clouds varies according to their importance. Indeed, sensitive applications require more than one region to avoid the risk of interruption. Given an efficient load balancing system to store the files in the cloud environment is a secure solution to guarantee an efficient dispersion of the stored files to avoid the most storage in one region. Consequently, damage to this region does not cause a loss of big data. For example, if there are many files that have a high-security level and these files are stored in the same region, a hack of the system damages all these high levels files. The load balancing ensures the dispersion of these files through different regions.

In the cloud environment, providers connect regions and availability zones through a point-to-point network. For economic, technical, and security reasons, cloud regions are geographically distributed at multiple points around the world. This means that the data passes from one region to another *via* the Internet network. Data travels through global routers all the time. The performance and the quality of the links in terms of speed and the level of security are not the same in all sections of the Internet network. Data availability of applications is an important factor for the cloud customer. The cloud infrastructure is designed to ensure this constraint. A global network of interconnected servers and systems provides nearly limitless fail-over scenarios. Cloud technology makes it possible to permanently replicate and synchronize any type of data (*Alzakholi et al., 2020*). In the event of a disruption server outage or network disruption and the cloud, setup will simply switch to a replica and prove to offer access to systems and data. For the end user, the transition is seamless in most scenarios, without realizing that a failure has occurred. Cloud security is a concern for any organization (*Singh & Chatterjee, 2017*). Security issues can arise when moving critical systems and sensitive data to a cloud computing solution (*Lee, 2013*). In addition, the load balancing problem is studied in several domains in literature. Indeed, the healthcare domain focused on the scheduling of the given quality reports to be treated by physicians in a hospital. The number of pages that have each report must be considered as a decision variable. this means that a load balancing of the total number of pages will be imposed (*Jemmali et al., 2022c*). On the other hand, in the domain of the industry, the maximization of the minimum completion time for the problem of the parallel machine is studied in *Jemmali & Alourani (2021)*. In this latter work, a mathematical model is proposed to solve the NP-hard problem. In the same domain, other work solves this problem approximately by proposing several heuristics and giving experimental results

resulting from these heuristics to compare between them (*Jemmali, Otoom & al Fayez, 2020*).

Several research projects studied load balancing in cloud computing. In *Al Nuaimi et al. (2012)* and *Ghomi, Rahmani & Qader (2017)*, the authors gave a survey regarding the application of load balancing in cloud computing. An analysis of the load balancing in cloud computing is detailed and discussed in *Sidhu & Kinger (2013)*.

The rest of the article is organized as follows. 'Literature review' is reserved for the literature review of the studied problem. 'Architecture and model' presents the novel architecture and model for the cloud system incorporating load balancing. The problem description is detailed in 'Problem description'. In 'Proposed algorithms', a presentation of the proposed algorithms is illustrated and discussed. The experimental results and the discussions of the obtained results are detailed in 'Experimental and discussion'. A conclusion and future directives are given in 'Conclusion'.

## LITERATURE REVIEW

According to the latest studies, cloud provider-side or client-side security measures are not sufficient. Researchers have to put a lot of effort into countering security threats and defending the cloud system in general. In *Arunarani, Manjula & Sugumaran (2017)*, the heights included a security service for task planning. For this, they developed an algorithm based on a hybrid optimization approach to minimize the risk rate. They claim that the processes developed can both minimize execution costs and meet time constraints. In *Chen et al. (2017)*, the authors developed new approaches to reduce monetary costs and use the cheapest resources. Their approach called SOLID consists of selective duplication of previous tasks and encryption of intermediate data. In *Fard, Prodan & Fahringer (2012)*, the authors introduced a new model for cloud pricing and a truthful scheduling mechanism. The goal of this work was to minimize the cost and the global execution time. Their results are compared with classical algorithms and Pareto solutions. In *Francis et al. (2018)*, the authors presented a summary of secure data flow planning models in the cloud environment. The article presents a solid mathematical study for the basic maintenance of dynamic nodes in graphs and which need updating for the base number of each vertex. Other researchers focused on edge-computing techniques in the cloud computing environment to enhance resource allocation and increase management quality (*Hua et al., 2019*). In addition, they affirmed that scheduling mechanisms could present better performance, especially in real-time applications. In this context, the authors in *Sang et al. (2022)* presented heuristics applied in device-edge-cloud cooperative computing. The authors considered that planning tasks enhanced the use of limited resources in the edge servers. They studied scheduling problems to find satisfaction and agreement between task numbers whose deadlines are met for cooperative computing at the device edge. Likewise, in *Wang et al. (2020)*, the authors developed a binary nonlinear programming (BNP) model to solve the problem of optimizing deadline violations in different heterogeneous computational environments like cloud, edge, *etc*. The goal is to maximize the number of completed tasks and enhance resource utilization. The authors in *Han et al. (2019)* proposed a general

model to solve task distribution and scheduling problems on edge networks in order to minimize the response time of these tasks. Indeed, jobs are generated in an arbitrary order and at arbitrary times on mobile devices. Then, they are unloaded on servers with upload and download delays. In the same context, the authors in *Aburukba, Landolsi & Omer (2021)* discussed the delay problem required by IoT applications. According to them, cloud computing generates unacceptable delays between IoT devices and cloud data centers. They preferred fog computing, which brings IT services closer to IoT devices. They developed heuristics based on a genetic algorithm to satisfy requests as much as possible within acceptable deadlines. In *Bezdan et al. (2021)*, the authors improved the search operator of traditional FPA by replacing the worst individuals with randomly generated new individuals in the search space to avoid getting stuck in local minima at the start of the optimization process. This improved FPA, called EEFPA, was used to find optimal scheduling of tasks in cloud computing environments, minimizing makespan as the primary goal. EEFPA was the best planner compared to similar approaches in this study. In the network, the load balancing is applied to schedule several packets to the different routers ensuring the load balancing of the total size of transmitted packets through routers (*Jemmali & Alquhayz, 2020a*). The gas turbine engine is another domain in that the load balancing is applied (*Jemmali et al., 2019*; *Jemmali, Melhim & Alharbi, 2019*). The Gray Wolf Optimizer (GWO) has been proposed for planning tasks in cloud computing to use resources more efficiently and minimize overall execution time (*Bacanin et al., 2019*). This algorithm has been compared with several scheduling methods such as FCFS, ACO, Performance Budget ACO (PBACO), and Min-Max algorithms. Experimental results showed that GWO was the best-performing scheduler and PBACO was his second-best. However, since the performance of his large-scale GWO has not been evaluated, it is not preferable when the number of tasks is large. In *Tawfeek et al. (2013)*, the authors developed an optimization of their colony of ants to handle task scheduling in cloud computing, with the aim of reducing makespan. This algorithm was compared with his two conventional algorithms such as FCFS and RR and showed better performance than both. The problem with this algorithm is that it converges slowly, requiring multiple iterations to get a usable solution. In *Hamad & Omara (2016)*, the authors proposed a genetic algorithm (GA)-based task scheduling algorithm to find the optimal assignment of tasks in cloud computing to optimize manufacturing margins and costs and resource utilization.

In *Jia, Li & Shi (2021)*, the authors presented a task scheduler based on an improved Whale Optimization Algorithm (IWOA). Standard WOA was improved with IWOA using two factors: nonlinear convergence coefficient and adaptive population size. IWOA outperformed the compared algorithms in terms of accuracy and convergence speed when planning small or large tasks in cloud computing environments.

Various partial computation offload algorithms have been designed for IoT systems in a heterogeneous 5G network. A review of this work shows that the algorithms were implemented for the purpose of minimizing energy consumption and reducing delay (*Singh et al., 2020*; *Yang et al., 2018*). However, researchers have paid little attention to the use of MEC load reduction for IoT security. In *Alladi et al. (2021)*, the authors implemented partial computation for many users uploading. The authors described a deep learning

engine (DLE)-intrusion detection architecture based on artificial intelligence (AI) to identify and classify media traffic in the Internet of Vehicles (IoV) into possible cyberattacks. These DLEs have also been deployed on MEC servers instead of the remote cloud. Taking into account the mobility of the vehicle and the real-time needs of the IoV network.

Rapid adoption and ease of use across all industries, the pervasiveness of the Internet of Things concept, and the continued development of infrastructure and technology have increased user demand for cloud computing, doubling data volumes and user demands. Scheduling tasks becomes a more difficult topic. Provisioning resources according to user requirements and maintaining the end-user quality of service (QoS) requirements is a daunting task (*Nayar, Ahuja & Jain, 2019*). The work that can be near our objective studied in this article is the load balancing of the size of files that must be stored in different storage support (*Alquhayz, Jemmali & Otoom, 2020*). In this work, the authors proposed different algorithms using different techniques like the iterative method and the probabilistic method. Recently, the authors in *Jemmali et al. (2022a)* apply the load balancing method by proposing several novel algorithms to solve the problem of the drone battery for monitoring the solar power plant. The proposed algorithms in this latter work are assessed and compared between them. The security parameters in scheduling data in clouds were considered in several works like *Meng et al. (2020)* and *Houssein et al. (2021)*.

Table 1 provides a scope of improvement of the related works discussed previously.

In the domain of smart parking, the number of persons in vehicles must be taken into consideration to schedule the vehicles on the available parking. Several algorithms are proposed to solve this problem based on the load balancing problem. This problem is proven to be NP-hard by the authors *Jemmali et al. (2022b)* and *Jemmali (2022)*. The budgeting and the management of the projects are exploited to solve a modeled problem of load balancing. In fact, in *Jemmali (2021b)*, the authors proposed heuristics to solve the problem of the scheduling of several projects characterized by their expected revenue. An experimental result shows the best-proposed heuristic in the work compared with all others. In the same context, the authors in *Jemmali (2021a)* proposed an optimal solution for the project assignment. Each project is characterized by its budget. The problem is to find a schedule that assigns all projects to the given municipalities ensuring the load balancing of the total budget in each municipality. Another work treated a similar problem in *Alharbi & Jemmali (2020)* and *Jemmali (2019)*.

In literature, scheduling problems are the subject of several types of research. Multiple models are developed (*Alharbi & Jemmali, 2020*; *Mahapatra, Dash & Pradhan, 2017*; *Jemmali, 2019*) to optimize the data transfer in different domains. The scheduling algorithms developed in *Melhim, Jemmali & Alharbi (2018)*, *al Fayez, Melhim & Jemmali (2019)*, *Alquhayz & Jemmali (2021b)*, *Jemmali, Melhim & Al Fayez (2022)*, *Hmida & Jemmali (2022)*, *Sarhan & Jemmali (2023)*, *Jemmali & Ben Hmida (2023)*, *Jemmali et al. (2022a)* and *Haouari, Gharbi & Jemmali (2006)* can be enhanced by the adoption of the proposed algorithms in this article.

The proposed algorithms can be applied to the problem described in *Boulila et al., (2010)*, *Driss et al. (2020)*, *Ghaleb et al. (2019)* and *Al-Sarem et al. (2020)*.

These existing works have several limitations that can be presented as follows:

**Table 1  Scope of improvement of previous work.**

| Num. | Scope of improvement | Reference |
|---|---|---|
| 1 | A security and cost-aware scheduling method for non-homogeneous jobs in a workflow run in the cloud | *Arunarani, Manjula & Sugumaran (2017)* |
| 2 | Task scheduling under selective-duplicate predecessor jobs to idle time and intermediate data encryption by exploiting tasks' laxity time | *Chen et al. (2017)* |
| 3 | Minimizing the completion time and monetary cost | *Fard, Prodan & Fahringer (2012)* |
| 4 | Survey of different previous works, by defining the factors required in securing workflows through the execution | *Francis et al. (2018)* |
| 5 | Algorithms to increase the parallelism and minimize the processing time | *Hua et al. (2019)* |
| 6 | Task scheduling problem to optimize the service level agreement: a new formulation into binary nonlinear programming and developed-heuristic with three stages | *Sang et al. (2022)* |
| 7 | Optimizing deadline violations for executing tasks: a formulation as a binary nonlinear programming model maximizing the number of completed tasks and optimizing the resource utilization of servers | *Wang et al. (2020)* |
| 8 | General model to solve task distribution and scheduling problems on edge networks in order to minimize the response time of these tasks | *Han et al. (2019)* |
| 9 | Delay problem required by IoT applications | *Aburukba, Landolsi & Omer (2021)* |
| 10 | Improved the search operator of traditional FPA by replacing the worst individuals with randomly generated new individuals in the search space to avoid getting stuck in local minima at the start of the optimization process | *Bezdan et al. (2021)* |
| 11 | Load balancing of the total size of transmitted packets through two routers | *Jemmali & Alquhayz (2020a)* |
| 12 | Load balancing in the gas turbine engine: algorithms and approximate solutions | *Jemmali et al. (2019)* and *Jemmali, Melhim & Alharbi (2019)* |
| 13 | Gray Wolf Optimizer (GWO) has been proposed for planning tasks in cloud computing to use resources more efficiently and minimize overall execution time | *Bacanin et al. (2019)* |
| 14 | An optimization of his colony of ants to handle task scheduling in cloud computing, with the aim of reducing makespan | *Tawfeek et al. (2013)* |
| 15 | A genetic algorithm (GA)-based task scheduling algorithm to find the optimal assignment of tasks in cloud computing to optimize manufacturing margins and costs and resource utilization | *Hamad & Omara (2016)* |
| 16 | Whale optimization algorithm to solve the task scheduling in cloud computing | *Jia, Li & Shi (2021)* |
| 17 | Various partial computation offload algorithms have been designed for IoT systems in a heterogeneous 5G network | *Singh et al. (2020)* and *Yang et al. (2018)* |
| 18 | Implementation of a partial computation for many users uploading | *Alladi et al. (2021)* |

**Table 1** (*continued*)

| Num. | Scope of improvement | Reference |
|------|---------------------|-----------|
| 19 | Provisioning resources according to user requirements and maintaining the end-user quality of service (QoS) requirements is a daunting task | *Nayar, Ahuja & Jain (2019)* |
| 20 | Dispatching-rules algorithms for the storage of files system: A load balancing | *Alquhayz, Jemmali & Otoom (2020)* |
| 21 | Load balancing method by proposing several novel algorithms to solve the problem of the drone battery | *Jemmali et al. (2022a)* |
| 22 | The security parameters in scheduling data in clouds | *Meng et al. (2020)* and *Houssein et al. (2021)* |

- Scalability: some algorithms cannot give a solution in an acceptable time for big-scale instances;
- Overhead: Different developed heuristics for load balancing can generate overhead;
- Limitation of implementation: Some heuristics can only be suitable to particular kinds of files and virtual machines with specific characteristics.

In this article, a novel method based on the grouping procedure is proposed. This method is applied to different scheduling routines and generates a set of algorithms that solve the studied problem. In *Alquhayz, Jemmali & Otoom (2020)*, the developed algorithms are based on the dispatching rules method. The proposed algorithms classify the files into different groups. The choice of files that contains different groups makes the schedule more dispersed and gives differentiated results. Changing the way that we select files into groups and between groups is the core of the difference between the proposed algorithms.

## ARCHITECTURE AND MODEL

In the cloud environment, most of the proposed models aim to minimize delay and increase performance. In our work, we propose a new model to assign planning data to appropriate regions by considering the file stability in each region.

The components of the model are as follows:

- Users are the workflow generators. Data can be files, databases, videos, *etc.*
- Scheduler represents the developed heuristics: The developed heuristics should provide suitable scheduling solutions which guarantee a minimum of makespan and the appropriate destination region. Heuristics consider incoming workflow, the queued data, and the resource allocation state.
- Cloud service provider: allocate adequate resources to the appropriate services, calculate costs, and guarantee the availability of the resources.
- Region 1 and 2: These are the cloud resources. It contains all available *VMs* that are capable of receiving storage data. Each region has its own characteristics like geographical position, cost, and availability parameters.

The main idea of this proposal is to assign data to a suitable cloud region. In the cloud, scheduling is an essential process for guiding files to be stored. After receiving user requests

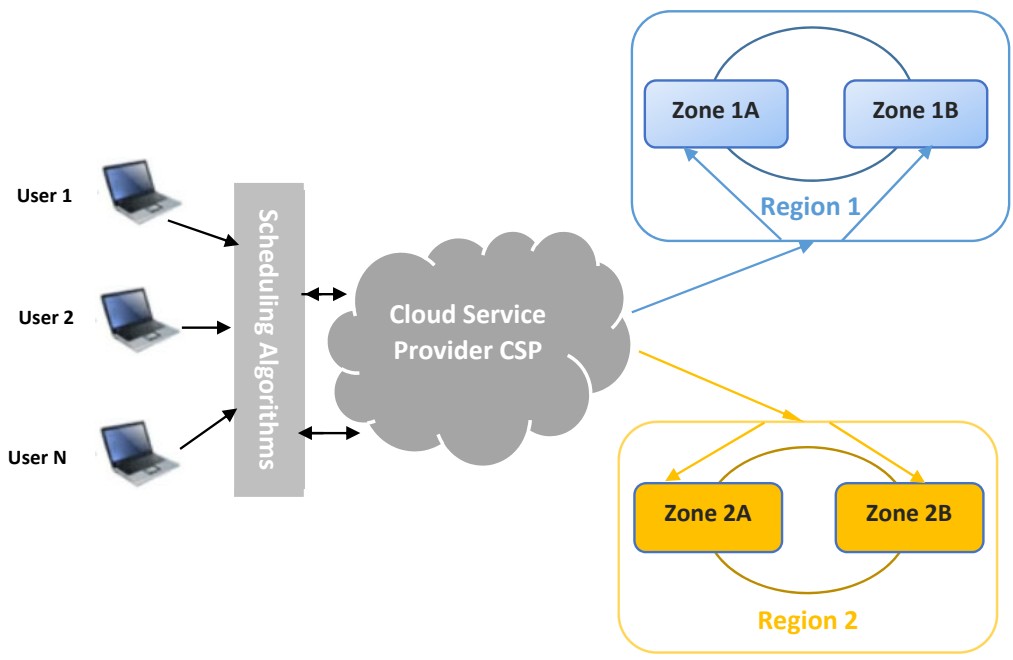

**Figure 1** **General overview of the load-balanced cloud environment for availability zones.**

and data, the scheduler component should gather accepted files, analyze them according to customer constraints and estimate the needed capacities.

Regularly, the cloud service provider sweeps up all available resources and collects information about regions. This component translates this information to the scheduler, which contains developed heuristics. The scheduler gathers information again, checks the developed heuristic results, and assigns each file to the suitable region. The scheduler component, the heart of our work, is working with the collaboration of a cloud service provider. It collects the necessary information and dynamically calculates the best solution to assign and dispatch each file into its corresponding region. Figure 1 shows the proposed model. In this section, a novel architecture and model for the studied problem will be presented and detailed.

## PROBLEM DESCRIPTION

In general, cloud providers do not charge for viewing or modifying data at the same level of infrastructure, but they charge for migrating data from one region to another. In addition to the cost of data transfer, migrating data from one region to another increases the risk of interception and hacking. The criteria for choosing regions and zones in the cloud environment is not only the cost but also the stability of the files in each region. Our goal in this work is to transfer files to the right places to avoid moving them several times. Moving and transferring files over the global network not only increases the cost of the cloud but also the likelihood of being intercepted, lost, hacked, *etc.*

**Table 2 Notations and their definitions.**

| Notations | Definitions |
|---|---|
| $FL$ | Set of files |
| $Vr$ | Set of virtual machines |
| $V_n$ | Number of available virtual machines |
| $F_n$ | Number of total files |
| $i$ | Index related to the file |
| $j$ | Index related to the virtual machine |
| $F_i$ | File number $i$ |
| $Vr_j$ | Virtual machine number $j$ |
| $Sz_i$ | Size of the file $F_i$ |
| $Tf_i$ | The used space when the file $F_i$ is executed |
| $Ts_j$ | The total used space in the virtual machine $Vr_j$ |
| $Ts_{min}$ | The minimum used space $\forall Vr_j, j = \{1, \ldots, V_n\}$ |

In our work, we will develop algorithms that provide load balancing between storage servers in different regions to minimize migration actions. The method that we propose ensures the fair distribution of several files to different virtual machines cited in different regions and zones.

In literature, several heuristics and algorithms are used to find an approximate solution (*Jemmali, 2021b*). *Alquhayz, Jemmali & Otoom (2020)*, *Jemmali (2021a)* prove that heuristics produce acceptable solutions in a cloud environment. These solutions may not be perfect, but they are still valuable.

All presented notations and their definitions are defined in Table 2.

Several objective functions can be adopted to solve the studied problem. In this article, we adopt the objective function detailed in Eq. (1). Hereafter, *Gfv* denotes the gap of the used space between the different virtual machines. This gap will be the objective that must be minimized.

$$Gfv = \sum_{j=1}^{V_n} (Ts_j - Ts_{min}). \tag{1}$$

The objective is to find a schedule that can minimize *Gfv*. In these circumstances, finding an approximate solution is very challenging. In this article, we propose several algorithms that solve approximately the studied problem. To give a clear idea of the studied problem, we give Example 1.

**Example 1** In this example, we give a scenario that can be realized in a real circumstance. Suppose that, there are three virtual machines and nine files to be executed by these virtual machines. In this case, we have $V_n = 3$ and $F_n = 9$. The objective is to find a schedule that can give an acceptable solution to assign all these files to different virtual machines. Table 3, illustrates the sizes of the different files. Assume that we will choose the shortest size-based algorithm. This algorithm is based on the following: we sort all files according to the increasing order of their size and the scheduling will be done one by one on the virtual machine that has the minimum value of $Ts_j$. The result

**Table 3  Nine files scenario with its different size.**

| $i$ | 1 | 2 | 3 | 4 | 5 | 6 | 7 | 8 | 9 |
|---|---|---|---|---|---|---|---|---|---|
| $Sz_i$ | 21 | 11 | 12 | 8 | 19 | 5 | 12 | 13 | 4 |

obtained by this algorithm is presented in Fig. 2. This Figure shows that in the virtual machine $Vr_1$ the files $\{2, 8, 9\}$ are executed. However, in the virtual machine $Vr_2$ the files $\{5, 6, 7\}$ are executed. Finally, in the virtual machine $Vr_3$ the files $\{1, 3, 4\}$ are executed. Now, the calculation of the $Ts_j$ is necessary to determine the total gap $Gfv$. As a result, the values of $Ts_1$, $Ts_2$, and $Ts_3$ are 28, 36, and 42, respectively. Consequently, $Gfv = \sum_{j=1}^{3}(Ts_j - Ts_{min} = (28 - 28) + (36 - 28) + (42 - 28)) = 22$. For this schedule, the gap between the virtual machines is 22. The objective is to find another schedule that gives a better result which means a gap of less than 22. Applying the algorithm of the longest size that sorts the files according to the decreasing order of their size and the scheduling will be done one by one on the virtual machine that has the minimum value of $Ts_j$. The result obtained by this algorithm is presented in Fig. 3. This Figure shows that in the virtual machine $Vr_1$ the files $\{1, 2, 9\}$ are executed. However, in the virtual machine $Vr_2$ the files $\{5, 6, 7\}$ are executed. Finally, in the virtual machine $Vr_3$ the files $\{3, 4, 8\}$ are executed. Now, the calculation of the $Ts_j$ is necessary to determine the total gap $Gfv$. As a result, the values of $Ts_1$, $Ts_2$, and $Ts_3$ are 36, 36, and 34, respectively. Consequently, $Gfv = \sum_{j=1}^{3}(Ts_j - Ts_{min} = (36 - 34) + (36 - 34) + (34 - 34)) = 4$. It is clear the first schedule presented in Fig. 1 gives a gap greater than the result obtained by schedule 2. The difference between these two schedules is $22 - 4 = 18$. So, just by changing the sorting method between the first algorithm and the second one we gain 18 units of gap value.

## PROPOSED ALGORITHMS

In this section, we present and detail all the proposed algorithms. These algorithms are based on the classification method. Indeed, the files are grouped into three groups. The choice of files that contains different groups makes the schedule more dispersed and gives differentiated results. Ten algorithms are proposed in this work. All these algorithms are based on the grouping method.

### New grouping method

This method subdivides the files into three groups $G_1$, $G_2$, and $G_3$. At the start, these groups are empty. The number of files in $G_1$, $G_2$, and $G_3$ are denoted by $f_1$, $f_2$, and $f_3$, respectively. In the practice, $f_1 = \frac{Fn}{3}$, $f_2 = \frac{Fn}{3}$, and $f_3 = Fn - f_1 - f_2$. Consequently, $G_1 = \{F_1, \ldots, F_{f_1}\}$, $G_2 = \{F_{f_1+1}, \ldots, F_{f_1+f_2}\}$, and $G_3 = \{F_{f_1+f_2+1}, \ldots, F_{Fn}\}$. It is clear that the groups $G_1$, $G_2$, and $G_3$ depend on the manner that initially the files are sorted. Therefore, the initial state of the set of files is very important to determine the groups. Ten algorithms are presented in this article based on the grouping method. Changing the way that we select files into groups and between groups is the core of the difference between the proposed algorithms. Figure 4 gives an example of the subdivision of the given set of files.

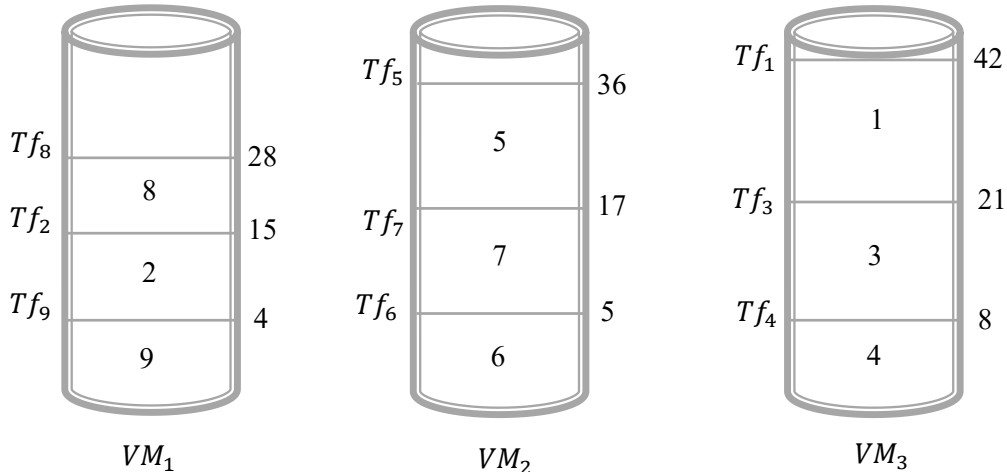

**Figure 2    Schedule 1 applying a shortest size-based algorithm for Example 1.**

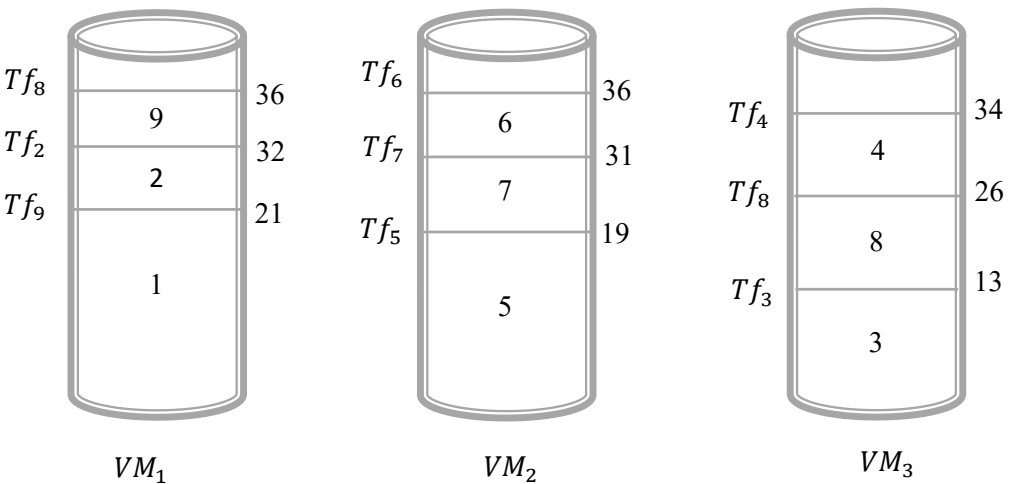

**Figure 3    Schedule 1 applying a longest size-based algorithm for Example 1.**

## Longest file size algorithm (*LFS*)

Firstly, the files are sorted according to the decreasing order of their size. The scheduling of the sorted files will be done one by one on the virtual machine that has the minimum value of $Tf_i$ until all files finish their execution. The complexity of this algorithm is depending on the algorithm that sorts the files. The heap sort is adopted for this algorithm. Consequently, the complexity of this algorithm is $O(nlogn)$. All these steps are described in Algorithm 2. Hereafter, we denoted by $DER(F)$ the procedure that receives as input a list of numbers $F$ and sorts these numbers in decreasing order. These numbers will be the sizes of the files that we want to sort them. The procedure $DER(F)$ is based on the heap sort method. Hereafter, we denoted by $SCL(F)$ the procedure that schedules the element $i$ ($\forall i, 1 \leq i \leq F_n$)

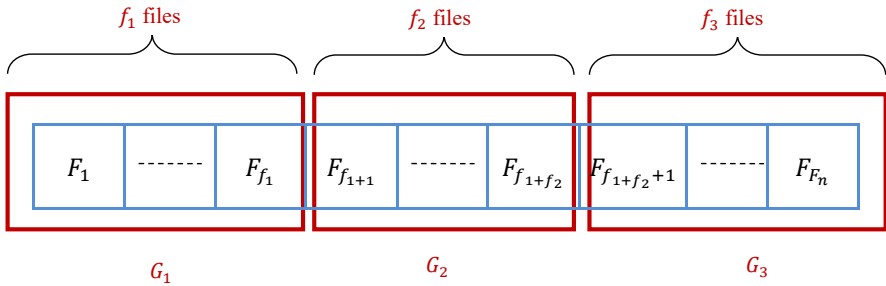

**Figure 4** Grouping of a set of files into three groups.

of $F$ on the virtual machine which has the minimum $Ts_j$. The instructions of $SCL(F)$ are detailed in Algorithm 1.

---

**Algorithm 1** Scheduling of a list $F$ Algorithm $(SCL(F))$

---

1: **for** $(i = 1$ to $F_n)$ **do**

2:      Determine $j$ subject to $\min\limits_{1 \leq j \leq V_n} Ts_j$

3:      Update $Ts_j = Ts_j + F_i$

4: **end for**

---

The instructions of $LFS$ are detailed in Algorithm 2.

---

**Algorithm 2** Longest file size Algorithm $(LFS)$

---

1: Call $DER(FL)$

2: Call $SCL(FL)$

3: Calculate $Gfv$

4: Return $Gfv$

---

## Third-grouped set algorithm (*TGS*)

The content of the groups depends on the manner that the files are sorted initially. We adopt three manners to sort the files.

- First manner: Take the files as given initially without applying any sorting.
- Second manner: Sort the files according to the increasing order of their size.
- Third manner: Sort the files according to the decreasing order of their size.

For each manner, firstly we create the groups $G_1$, $G_2$, and $G_3$. After that, we constitute a permutation for these groups. There are six possibilities to constitute a sequence of groups. The first sequence is $G_1$, $G_2$, and $G_3$ denoted as $\{G_1, G_2, G_3\}$. For this sequence, we schedule all files in $G_1$, next we schedule all files in $G_2$ and finally, we schedule all files in $G_3$. The second sequence is $\{G_1, G_3, G_2\}$. The third sequence is $\{G_2, G_1, G_3\}$. The fourth sequence is $\{G_2, G_3, G_1\}$. The fifth sequence is $\{G_3, G_1, G_2\}$. The last sequence is $\{G_3, G_2, G_1\}$. So, for each manner, six sequences are executed and the best solution is picked and returned.

### Third-grouped with minimum-load algorithm (*TGM*)

The division into groups of files is adopted in this algorithm. Three groups are created by the same method described in the section 'New grouping method' with the three manners detailed in the same subsection. For each manner, a solution is calculated and the best solution is selected. The algorithm is designed in five steps. The first step is to apply the first manner and create the three groups $G_1$, $G_2$, and $G_3$. The second step is to calculate the load of each group. The load is the sum of all sizes of the files in the group. Indeed, the load of $G_1$ is denoted by $Lo_1$ and equal to $\sum_{i=1}^{f_1}(Sz_i)$. The load of $G_2$ and $G_3$ are denoted by $Lo_2$ and $Lo_3$, respectively. So, we have $Lo_2 = \sum_{i=f_1+1}^{f_1+f_2}(Sz_i)$ and $Lo_3 = \sum_{i=f_1+f_2+1}^{F_n}(Sz_i)$. The third step is to choose the group which has the minimum load. This group will be denoted by $Gc$. The fourth step is to schedule the first file in $Gc$. We update loads of different groups and a new choice of a $Gc$ will be determined and so on until the schedule of all the files. The total gap is calculated and denoted by $Gfv_1$. We restart step 1 with the second manner described in the above Subsection and the total gap for this solution is calculated and denoted by $Gfv_2$. Finally, for the fifth step, we restart, step 1 with the application of the third manner, and a new gap is calculated and denoted by $Gfv_3$. The best solution $Gfv$ is calculated as given in Eq. (2).

$$Gfv = \min_{1 \le k \le 3}(Gfv_k) \tag{2}$$

### Third-grouped excluding-files with minimum-load algorithm (*TEM*)

The first step of this algorithm is the select the $V_n$ longest files. Each file will be scheduled in a distinguished virtual machine. Now, the $F_n - V_n$ remaining files will be scheduled in the virtual machines according to *TGM*. We denoted by $EXL(F)$ the function that returns a list that contains the $V_n$ longest files among $F$. We denoted by $Rem(F)$ the function that returns a list that contains the $F_n - V_n$ remaining files after excluding the $V_n$ longest files among $F$. Hereafter, we denoted by $IER(F)$ the procedure that receives as input a list of numbers $F$ and sorts these numbers in decreasing order. Hereafter, we denoted by $GrP(F)$ the procedure that subdivided the listed files $F$ into the three groups $G_1$, $G_2$, and $G_3$ described in the section 'New grouping method'. The procedure $MG()$ is responsible to return the group that has the maximum load. The procedure $SCLF(L)$ is responsible to schedule the first element of $L$ on the available virtual machine. All steps of the algorithm are described in the Algorithm 3.

### Third-grouped one-by-one algorithm (*TGO*)

The determination of the three groups described above is adopted for this algorithm. These three groups will be created as described in the section 'New grouping method'. The three manners as also applied. For each manner, firstly we create the groups $G_1$, $G_2$, and $G_3$. After that, we constitute a permutation for these groups. There are six possibilities to constitute the order of groups. The first order is $G_1$, $G_2$, and $G_3$ and denoted by $Order_1$. This means that we schedule the first file from $G_1$, the first file from $G_2$, and the first file from $G_3$. For the fifth order, we schedule applying the same method, the first file from

**Algorithm 3** Third-Grouped Excluding-Files with Minimum-Load Algorithm (*TEM*)

1: Set $Ls = EXL(FL)$
2: Call $SCL(Ls)$
3: Set $Lr = Rem(FL)$
4: **for** (*manner* $= 1$ to 3) **do**
5:     **if** (*manner* $= 2$) **then**
6:         Call $DER(Lr)$
7:     **end if**
8:     **if** (*manner* $= 3$) **then**
9:         Call $IER(Lr)$
10:     **end if**
11:     Call $GrP(Lr)$
12:     **for** ($i = 1$ to $F_n - V_n$) **do**
13:         Calculate $Lo_1, Lo_2, Lo_3$
14:         Set $Gm = MG()$
15:         Call $SCLF(Gm)$
16:         Update $Gm$
17:     **end for**
18:     Calculate $Gfv_{manner}$
19: **end for**
20: Calculate $Gfv = \min_{1 \leq manner \leq 3} (Gfv_{manner})$
21: Return $Gfv$

each group, following the order of the group, is scheduled. The second order is $\{G_1, G_3, G_2\}$ and denoted by $Order_2$. The third order is $\{G_2, G_1, G_3\}$ and denoted by $Order_3$. The fourth order is $\{G_2, G_3, G_1\}$ and denoted by $Order_4$. The fifth order is $\{G_3, G_1, G_2\}$ and denoted by $Order_5$. The last order is $\{G_3, G_2, G_1\}$ and denoted by $Order_6$. So, for each manner, six orders are executed and the best solution is picked and returned. We denoted by $SCG(Order_h)$ with $h = \{1, \ldots, 6\}$ the procedure that schedules the files following the order received as input until scheduling of all files.

### Three-files swap third-grouped algorithm (*TST*)

The determination of the three groups described above is adopted for this algorithm. These three groups will be created as described in the section 'New grouping method'. The three manners as also applied. For each manner, firstly we create the groups $G_1$, $G_2$, and $G_3$. After that, we constitute a permutation for these groups. There are six possibilities to constitute the order of groups as described in the section 'Third-grouped one-by-one algorithm (*TGO*)'. So, for each manner, six orders are executed. For each order, a swap of three files is applied. These files are the first file $F1$ from $G_1$, the first file $F2$ from $G_2$, and the first file $F3$ from $G_3$. The swapping is as follows:

- Restore a copy of $F1$

---

**Algorithm 4** Third-Grouped One-by-one Algorithm (*TGO*)

---

1: **for** (*manner* $= 1$ to 3) **do**
2:      **if** (*manner* $= 2$) **then**
3:          Call *DER*(*FL*)
4:      **end if**
5:      **if** (*manner* $= 3$) **then**
6:          Call *IER*(*FL*)
7:      **end if**
8:      Call *GrP*(*FL*)
9:      **for** ($h = 1$ to 6) **do**
10:          Call *SCG*($Order_h$)
11:          Calculate $Gfv^h_{manner}$
12:      **end for**
13: **end for**
14: Calculate $Gfv = \min\limits_{1 \leq h \leq 6} (\min\limits_{1 \leq manner \leq 3} (Gfv^h_{manner}))$
15: Return *Gfv*

---

- Apply a translation of the $f_1 - 1$ files to the left beginning with position 2 and ending with position $f_1$.
- Move $F2$ at the end of $G_1$.
- Move $F3$ at the front of $G_2$.
- Move the stored copy of $F1$ at the front of $G_3$.

Now, after the swapping *TGO* described in the 'Third-grouped one-by-one algorithm (*TGO*)' on the new set of files obtained after swapping.

Hereafter, the procedure *SWap*($L1, L2, L3$) is responsible to swap $F1$, $F2$, and $F3$ the first files of the lists $L1, L2$, and $L3$ given as input, as the description above.

## On-tenth-files swap third-grouped algorithm (*OST*)

The determination of the three groups described above is adopted for this algorithm. These three groups will be created as described in the section 'New grouping method'. The three manners as also applied. For each manner, firstly we create the groups $G_1$, $G_2$, and $G_3$. After that, we constitute a permutation for these groups. There are six possibilities to constitute the order of groups as described in the section 'Third-grouped one-by-one algorithm (*TGO*)'. So, for each manner, six orders are executed. For each order, a swap of three files is applied. These files are the first $\frac{F_n}{10}$ files from $G_1$, the first $\frac{F_n}{10}$ files from $G_2$, and the first $\frac{F_n}{10}$ files from $G_3$. The swapping is as follows:

- Restore a copy of the first $\frac{F_n}{10}$ files from $G_1$
- Apply a translation of the $f_1 - 10$ files to the left beginning with position 11 and ending with position $f_1$.
- Move the first $\frac{F_n}{10}$ files from $G_2$ at the end of $G_1$.
- Move the first $\frac{F_n}{10}$ files from $G_3$ at the front of $G_2$.
- Move the stored copy of the first $\frac{F_n}{10}$ files from $G_1$ at the front of $G_3$.

---

**Algorithm 5** Three-files Swap Third-Grouped Algorithm (*TST*)

---

1: **for** (*manner* = 1 to 3) **do**
2:     **if** (*manner* = 2) **then**
3:         Call *DER*(*FL*)
4:     **end if**
5:     **if** (*manner* = 3) **then**
6:         Call *IER*(*FL*)
7:     **end if**
8:     Call *GrP*(*FL*)
9:     Call *SWap*($G_1, G_2, G_3$)
10:     **for** (*h* = 1 to 6) **do**
11:         Call *SCG*(*Order$_h$*)
12:         Calculate $Gfv^h_{manner}$
13:     **end for**
14: **end for**
15: Calculate $Gfv = \min\limits_{1 \le h \le 6} (\min\limits_{1 \le manner \le 3} (Gfv^h_{manner}))$
16: Return *Gfv*

---

Now, after the swapping *TGO* described in the 'Third-grouped one-by-one algorithm (*TGO*)' on the new set of files obtained after swapping.

## Best-value of three algorithms (*BVT*)

This algorithm returns the minimum value after running the *LFS*, *TGS*, and *OST*.

---

**Algorithm 6** Best-value of three algorithms (*BVT*)

---

1: Call *LFS*
2: Set $Gfv_1 = Gfv$
3: Call *TGS*
4: Set $Gfv_2 = Gfv$
5: Call *OST*
6: Set $Gfv_3 = Gfv$
7: Calculate $Gfv = \min\limits_{1 \le h \le 3} (Gfv_h)$
8: Return *Gfv*

---

## Best-value of five algorithms (*BFI*)

This algorithm returns the minimum value after running the *LFS*, *TGM*, *TEM*, *TGO*, and *TST*.

## Best-value of four algorithms (*BFO*)

This algorithm returns the minimum value after running the *LFS*, *TGS*, *TST*, and *OST*.

Based on the above algorithms, it is clear to see that the algorithm *BVT* dominates the three used algorithms *LFS*, *TGS*, and *OST*. In the same context, the algorithm *BVF* dominates the five used algorithms *LFS*, *TGM*, *TEM*, *TGO*, and *TST*. Finally, the algorithm

**Table 4  Choice of $(F_n, V_n)$.**

| $F_n$ | $V_n$ |
|---|---|
| 12,32,52 | 4,5,6 |
| 60,160,260,360 | 4,6,8,11 |
| 450,550,650 | 6,8,11 |

*BVF* dominates the four used algorithms. Consequently, we discussed only six algorithms in the experimental results. These algorithms are *LFS*, *TGS*, *OST*, *BVT*, *BFI*, and *BFO*.

## EXPERIMENTAL AND DISCUSSION

The performance of the proposed algorithms is measured and discussed in this section. Several classes of instances are coded and tested. These instances and the proposed algorithms are codded in C++ using a computer with an i5 processor and memory of 8 GB. The proposed procedures are tested on a set of instances that are detailed in the following Subsection.

### Instances

The tested instances are coded to be used by the proposed algorithms measuring the performance in terms of gap and time. These instances are depending on the manner that we generate the $Sz_i$ values. Indeed, the generation of $Sz_i$ is based on two distributions. The first one is the uniform distribution and is denoted by $UN[.]$. The second one is the normal distribution and is denoted by $NO[.]$.

The generated classes are illustrated as follows:

- *Class* 1: $Sz_i$ in $UN[25, 130]$.
- *Class* 2: $Sz_i$ in $UN[110, 370]$.
- *Class* 3: $Sz_i$ in $NO[220, 25]$.
- *Class* 4: $Sz_i$ in $NO[330, 110]$.

The choice of the number of virtual machines and the number of files that can be tested are presented in Table 4.

For each number of virtual machines and each number of files, 10 different instances were generated. In total, the number of generated instances is $(3 \times 3 + 4 \times 4 + 3 \times 3) \times 10 \times 4 = 1,360$. The generation of instances is inspired by the analysis presented in *Alquhayz, Jemmali & Otoom (2020)*; *Alquhayz & Jemmali (2021a)*, and *Jemmali & Alquhayz (2020b)*.

### Metrics

All algorithms presented in 'Proposed algorithms' will be discussed based on several metrics. These metrics are defined as follows.

- $\overrightarrow{Z}$ The minimum value obtained after executing of all algorithms.
- $Z$ The value of the presented algorithm.
- $Mp$ The percentage of instances when $\overrightarrow{Z} = Z$.
- $Gp = \frac{Z - \overrightarrow{Z}}{Z}$, if $Z = 0$, then $Gp = 0$.

**Table 5   Overall results for all algorithms.**

|        | LFS    | TGS    | OST    | BVT    | BFI    | BFO     |
|--------|--------|--------|--------|--------|--------|---------|
| Mp     | 29.2%  | 61.3%  | 40.6%  | 79.3%  | 56.4%  | **86.5%** |
| Ag     | 0.332  | 0.115  | 0.286  | 0.044  | 0.195  | **0.021** |
| Time   | .      | 0.0006 | 0.0006 | 0.0013 | 0.0021 | 0.0018  |

Notes.
Bold indicates the best results and the underline indicates the second-best results.

- *Ag* The average *Gp* for a fixed set of instances.
- *Time* The time of execution of an algorithm for a fixed set of instances. This time is in seconds and we recorded it as "." if the time is less than 0.0001 s.

## Discussion results

In this subsection, we discuss the performance of the proposed algorithms. This discussion is based on five kinds of analyses. The first kind is an overall analysis of the obtained results. The second kind is based on the number of files discussed. The third kind is based on the number of virtual machines discussed. While the fourth kind is based on the class's discussion. Finally, the fifth kind is based on the pair discussion. These kinds of analyses are discussed separately in the following subsections.

### Overall results

Table 5 presents the overall results for all algorithms. This table shows that the best algorithm is *BFO* in 86% of cases with an average gap of 0.021 and an average running time of 0.0018 s. The second best algorithm is *BVT* in 79.3% of cases with an average gap of 0.044 and an average running time of 0.0013 s. Table 5 shows that the maximum average gap of 0.332 is obtained by the *LFS* algorithm. The minimum running time of 0.0006 s is reached for *TGS* and *OST* algorithms. While the average running time of the *LFS* algorithm is less than 0.0001 s.

### Number of files discussion

In this subsection, we discuss the variation of the average gap and time when the number of files changes. Table 6 presents the average gap $Gfv$ of all algorithms according to the number of files $F_n$. This latter table shows that for the best algorithm, *BFO* the minimum average gap of 0.001 is reached when $F_n = 60$. The second minimum value of the average gap of 0.002 is obtained when $F_n = 360$. On the other hand, for *BFO*, the maximum average gap of 0.054 is obtained when $F_n = 550$.

Table 7 presents the average running time *Time* in seconds of all algorithms according to the number of files $F_n$. This latter table shows that the maximum average running time of 0.0048 is reached when $F_n = 550$ for algorithm *BFO*. While the minimum average running time of less than 0.0001 is reached four times for the *LFS* algorithm when $F_n = \{12, 32, 52, 60\}$.

**Table 6** The average gap *Gfv* of all algorithms according to the number files $F_n$.

| $F_n$ | LFS | TGS | OST | BVT | BFI | BFO |
|---|---|---|---|---|---|---|
| 12 | 0.134 | 0.016 | 0.016 | 0.016 | 0.083 | **0.011** |
| 32 | 0.355 | 0.063 | 0.252 | 0.022 | 0.259 | **0.020** |
| 52 | 0.253 | 0.203 | 0.238 | 0.067 | 0.081 | **0.015** |
| 60 | 0.328 | 0.052 | 0.300 | 0.008 | 0.239 | **0.001** |
| 160 | 0.264 | 0.166 | 0.432 | 0.049 | 0.139 | **0.042** |
| 260 | 0.277 | 0.102 | 0.283 | 0.035 | 0.221 | **0.033** |
| 360 | 0.434 | 0.065 | 0.252 | 0.003 | 0.241 | **0.002** |
| 450 | 0.474 | 0.218 | 0.344 | 0.151 | 0.214 | **0.006** |
| 550 | 0.414 | 0.118 | 0.490 | 0.078 | 0.154 | **0.054** |
| 650 | 0.368 | 0.147 | 0.193 | 0.023 | 0.305 | **0.023** |

Notes.
Bold indicates the best results and the underline indicates the second-best results.

**Table 7** The running time *Time* of all algorithms according to the number files $F_n$.

| $F_n$ | LFS | TGS | OST | BVT | BFI | BFO |
|---|---|---|---|---|---|---|
| 12 | . | 0.0001 | . | 0.0001 | 0.0002 | 0.0002 |
| 32 | . | 0.0002 | 0.0001 | 0.0003 | 0.0002 | 0.0003 |
| 52 | . | 0.0003 | 0.0002 | 0.0004 | 0.0005 | 0.0005 |
| 60 | . | 0.0002 | 0.0002 | 0.0005 | 0.0008 | 0.0006 |
| 160 | **0.0001** | 0.0004 | 0.0003 | 0.0008 | 0.0014 | 0.0011 |
| 260 | **0.0002** | 0.0007 | 0.0006 | 0.0011 | 0.0021 | 0.0017 |
| 360 | **0.0001** | 0.0006 | 0.0008 | 0.0015 | 0.0026 | 0.0022 |
| 450 | **0.0001** | 0.0010 | 0.0014 | 0.0025 | 0.0039 | 0.0038 |
| 550 | **0.0001** | 0.0012 | 0.0020 | 0.0033 | 0.0046 | 0.0048 |
| 650 | **0.0001** | 0.0013 | 0.0013 | 0.0027 | 0.0047 | 0.0040 |

Notes.
Bold indicates the best results and the underline indicates the second-best results.

**Table 8** The average gap *Gfv* of all algorithms according to the number of virtual machines $V_n$.

| $V_n$ | LFS | TGS | OST | BVT | BFI | BFO |
|---|---|---|---|---|---|---|
| 4 | 0.324 | 0.031 | 0.424 | 0.007 | 0.259 | **0.000** |
| 5 | 0.279 | 0.132 | 0.147 | 0.061 | 0.109 | **0.025** |
| 6 | 0.371 | 0.146 | 0.244 | 0.063 | 0.196 | **0.056** |
| 8 | 0.355 | 0.075 | 0.260 | 0.012 | 0.234 | **0.006** |
| 11 | 0.273 | 0.176 | 0.287 | 0.073 | 0.130 | **0.004** |

Notes.
Bold indicates the best results and the underline indicates the second-best results.

### Number of virtual machines discussion

In this subsection, we discuss the variation of the average gap and time when the number of virtual machines changes. Table 8 presents the average gap *Gfv* of all algorithms according to the number of virtual machines $V_n$.

**Table 9** The average time *Time* of all algorithms according to the number of virtual machines $V_n$.

| $V_n$ | LFS | TGS | OST | BVT | BFI | BFO |
|---|---|---|---|---|---|---|
| 4 | . | 0.0003 | 0.0002 | 0.0006 | 0.0010 | 0.0008 |
| 5 | . | 0.0002 | 0.0001 | 0.0003 | 0.0003 | 0.0003 |
| 6 | **0.0001** | 0.0006 | 0.0006 | 0.0012 | 0.0019 | 0.0017 |
| 8 | **0.0001** | 0.0008 | 0.0009 | 0.0017 | 0.0029 | 0.0026 |
| 11 | **0.0001** | 0.0009 | 0.0011 | 0.0021 | 0.0032 | 0.0031 |

Notes.
Bold indicates the best results and the underline indicates the second-best results.

**Table 10** The average gap *Gfv* of all algorithms according to classes.

| Class | LFS | TGS | OST | BVT | BFI | BFO |
|---|---|---|---|---|---|---|
| 1 | 0.405 | 0.176 | 0.348 | 0.036 | 0.312 | **0.002** |
| 2 | 0.347 | 0.157 | 0.472 | 0.056 | 0.222 | **0.020** |
| 3 | 0.248 | 0.053 | 0.145 | 0.039 | 0.096 | **0.029** |
| 4 | 0.329 | 0.074 | 0.179 | 0.045 | 0.150 | **0.031** |

Notes.
Bold indicates the best results and the underline indicates the second-best results.

**Table 11** The average running time *Time* in seconds of all algorithms according to classes.

| Class | LFS | TGS | OST | BVT | BFI | BFO |
|---|---|---|---|---|---|---|
| 1 | **0.0001** | 0.0006 | 0.0006 | 0.0012 | 0.0021 | 0.0018 |
| 2 | **0.0001** | 0.0006 | 0.0006 | 0.0013 | 0.0021 | 0.0019 |
| 3 | . | 0.0005 | 0.0007 | 0.0013 | 0.0020 | 0.0019 |
| 4 | . | 0.0006 | 0.0006 | 0.0012 | 0.0020 | 0.0018 |

Notes.
Bold indicates the best results and the underline indicates the second-best results.

Table 9 presents the average time *Time* of all algorithms according to the number of virtual machines $V_n$.

### Classes discussion

In this subsection, we discuss the variation of the average gap and the time when the class changes.

Table 10 presents the average gap *Gfv* of all algorithms according to the classes. This latter table shows that the minimum average gap of 0.002 is obtained for *BFO* and for Class 1. While the class that has the maximum average gap for the *BFO* algorithm is Class 4. Regarding all algorithms, the maximum average gap of 0.472 is obtained for the *OST* algorithm and for Class 2. We can see that, for the BFO algorithm, Class 1 is easier than the others because for this class the average gap is the minimum value of 0.002.

Table 11 presents the average gap running time *Time* of all algorithms according to the classes.

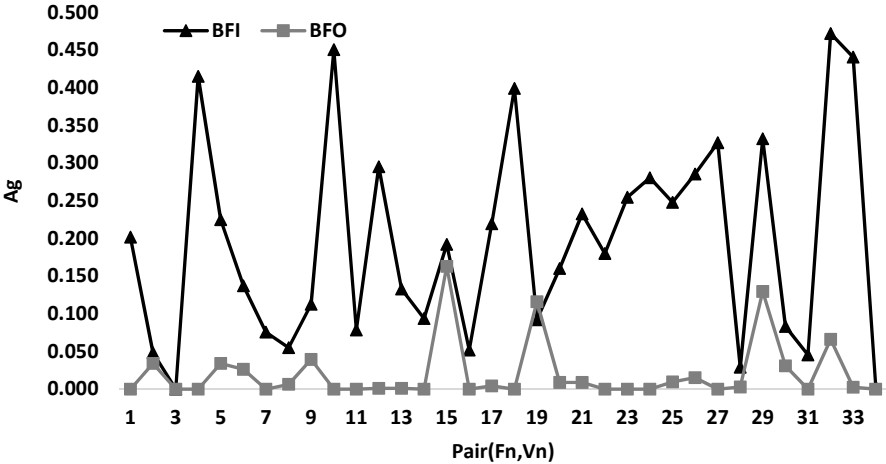

**Figure 5 Comparison between the best algorithm *BFO* and *BFI* according to pair ($F_n$, $V_n$).**

### Pair discussion

In this subsection, we discuss the variation of the average gap when the pair of ($F_n$, $V_n$) changes.

Figure 5 presents a comparison between the best algorithm *BFO* and *BFI* according to the pair($F_n$, $V_n$). This latter figure shows that the curve of *BFO* is always below the curve of *BFI* for all values of the pairs ($F_n$, $V_n$). This explains that *BFI* is the best algorithm. In addition, it is easy to see that the total number of the different pairs is 34.

### Comparison to existing algorithms

In this subsection, we discuss the comparison between the proposed algorithms and the existing ones. In the literature, in *Alquhayz, Jemmali & Otoom (2020)*, the authors develop algorithms to solve the storage problem. The best three algorithms in the latter work are *NISA*, *SIDA$^r$*, and *SDIA$^r$* with percentages of 45.2%, 75.2%, and 41%, respectively as detailed in Table 7 in *Alquhayz, Jemmali & Otoom (2020)*. On the other hand, the three best-proposed algorithms are *TGS*, *BVT*, and *BFO* with percentage of 61.3%, 79.3%, and 86.5%, respectively. Now, we compare the three best algorithms in *Alquhayz, Jemmali & Otoom (2020)* to *TGS*, *BVT*, and *BFO*. Hereafter, the percentage is calculated based on the minimum value obtained over the six algorithms (three best-existing algorithms and three best-proposed ones).

Table 12 presents an overall comparison between the three existing best algorithms and the three best-proposed algorithms. This table shows that the best algorithm is *BFO* in 63.2% of cases with an average gap of 0.145 and an average running time of 0.0018 s. The second best algorithm is *BVT* in 58.2% of cases with an average gap of 0.167 and an average running time of 0.0013 s. Table 12 shows that the maximum average gap of 0.429 is obtained by the *NISA* algorithm. This table shows that the proposed algorithms outperform those developed in the literature. The best existing algorithm is *SIDA$^r$* with

**Table 12** Overall comparison between the three existing best algorithms and the three best-proposed algorithms.

|     | NISA    | SIDAʳ   | SDIAʳ   | TGS     | BVT     | BFO     |
|-----|---------|---------|---------|---------|---------|---------|
| Mp  | 21.8%   | 50.8%   | 34.3%   | 47.3%   | 58.2% | **63.2%** |
| Ag  | 0.429   | 0.232   | 0.361   | 0.231   | 0.167 | **0.145** |
| Ag  | **0.0001** | 0.0019 | 0.0017 | 0.0006 | 0.0013 | 0.0018 |

**Notes.**
Bold indicates the best results and the underline indicates the second-best results.

**Table 13** Comparison of the average gap *Gfv* values between the three existing best algorithms and the three best-proposed algorithms according to the number files $F_n$.

| $F_n$ | NISA  | SIDAʳ    | SDIAʳ    | TGS      | BVT      | BFO      |
|-------|-------|----------|----------|----------|----------|----------|
| 12    | 0.130 | 0.113    | 0.244    | 0.011 | 0.011 | **0.006** |
| 32    | 0.399 | 0.269    | 0.365    | 0.118    | 0.078 | **0.077** |
| 52    | 0.360 | **0.140** | 0.359   | 0.322    | 0.202    | 0.154 |
| 60    | 0.452 | 0.207    | 0.366    | 0.195    | 0.155 | **0.150** |
| 160   | 0.500 | **0.150** | 0.363   | 0.446    | 0.352    | 0.349 |
| 260   | 0.475 | 0.274 | **0.258** | 0.337 | 0.280    | 0.274 |
| 360   | 0.467 | 0.122    | 0.497    | 0.101 | **0.041** | **0.041** |
| 450   | 0.486 | 0.328    | 0.483    | 0.230    | 0.163 | **0.018** |
| 550   | 0.590 | 0.435    | **0.249** | 0.342   | 0.316    | 0.299 |
| 650   | 0.357 | 0.341    | 0.385    | 0.134    | 0.010 | **0.009** |

**Notes.**
Bold indicates the best results and the underline indicates the second-best results.

a percentage of 50.8%, while the best-proposed algorithm is *BFO* with a percentage of 63.2%.

Table 13 presents the comparison of the average gap *Gfv* values between the three existing best algorithms and the three best-proposed algorithms according to the number files $F_n$. This latter table shows that for the best algorithm, *BFO* the minimum average gap of 0.006 is reached when $F_n = 12$. The second minimum value of the average gap of 0.009 is obtained when $F_n = 650$. On the other hand, for *BFO*, the maximum average gap of 0.349 is obtained when $F_n = 160$. Seven times *BFO* reach the minimum average gap values when $F_n = \{12, 32, 60, 260, 360, 450, 650\}$.

The proposed algorithms show their efficiency in the average gap. Indeed, the minimum average gap of 0.021 is reached when comparing all the proposed algorithms and the minimum average gap of 0.145 when comparing the existing algorithms to the proposed ones. The proposed algorithms are non-dominant. This means that the permutation of some tuples of these algorithms can give better results. The three best-proposed algorithms are *TGS*, *BVT*, and *BFO*. The results detailed in tables and figures show the performance of the algorithms. The application of the grouping method has a remarkable impact on the performance of the algorithms. Indeed, these three algorithms are based on the grouping method. This means that the grouping method shows its efficiency in the studied problem.

## CONCLUSION

In this article, a developed optimized algorithms scheduling based on load balancing for minimizing data migration from one region to another in a cloud environment was presented. The concept and the model are presented and explained. A novel grouping method is presented. This method is utilized to obtain performed algorithms to solve the studied problem. Ten algorithms are proposed. Due to the dominance rule between the algorithms, only six algorithms are discussed in the experimental results. Four classes of instances are generated and tested. These classes resulted in 1,360 instances in total. These experimental results show that the best algorithm is the "Best-value of four algorithms (*BFO*)" in 86.5% of cases with an average gap of 0.021 and an average running time of 0.0018 s. Cloud security is the first challenge for developers and researchers. For a company, choosing the best region to keep sensitive data is an important task because it avoids unnecessary migration from one region to another and this can decrease security levels and increase risks. By giving initial solutions, our proposed algorithms can be enhanced and give better solutions. In the future, the performance of our algorithms can be evaluated in the case of big data flow. The proposed algorithms can be enhanced by applying some meta-heuristics.

### Funding

This work was supported by the Deputyship for Research & Innovation, Ministry of Education in Saudi Arabia through the project number (IFP-2022-34). The funders had no role in study design, data collection and analysis, decision to publish, or preparation of the manuscript.

### Grant Disclosures

The following grant information was disclosed by the authors:
The Deputyship for Research & Innovation, Ministry of Education in Saudi Arabia: IFP-2022-34.

### Competing Interests

The authors declare there are no competing interests.

### Author Contributions

- Sarah Eljack conceived and designed the experiments, analyzed the data, prepared figures and/or tables, authored or reviewed drafts of the article, and approved the final draft.
- Mahdi Jemmali conceived and designed the experiments, performed the experiments, performed the computation work, authored or reviewed drafts of the article, and approved the final draft.
- Mohsen Denden analyzed the data, performed the computation work, prepared figures and/or tables, and approved the final draft.

- Sadok Turki conceived and designed the experiments, prepared figures and/or tables, and approved the final draft.
- Wael M. Khedr performed the experiments, prepared figures and/or tables, and approved the final draft.
- Abdullah M. Algashami analyzed the data, prepared figures and/or tables, authored or reviewed drafts of the article, and approved the final draft.
- Mutasim ALsadig analyzed the data, prepared figures and/or tables, and approved the final draft.

### Data Availability

The class of instances used in the experimental results and code are available in the Supplemental Files.

### Supplemental Information

Supplemental information for this article can be found online at http://dx.doi.org/10.7717/peerj-cs.1513#supplemental-information.

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
