# Peer review of "A secure solution based on load-balancing algorithms between regions in the cloud environment"

_PeerJ Computer Science, doi:10.7717/peerj-cs.1513_

## Round 0.1 · original submission · Major Revisions

According to the reviewer's comments, you can resubmit the manuscript with major revisions.

-Best regards,
Ahyoung.

·

Basic reporting

In this paper, the authors have presented a novel model regarding the regional security and load balancing of files in the cloud environment. The authors have presented seven algorithms. They have claimed that there is no dominance between these algorithms. However, the paper needs the following modifications before its possible inclusion in this esteemed journal.

- Introduction: Lack of references to other research papers.
- Load balancing is a well-known problem. However, "secure solution" needs to be justified in the Abstract as well as Introduction.
- Add a comparison table with the scope of improvement of each paper in the literature review section.
- The first paragraph of "architecture and model" should be moved to "literature review"
- The number of regions should be increased. The popular cloud simulator cloudsim/cloudanalyst is containing six regions and Amazon EC2 also contains more regions.
- Figure 1: Zone or Availability Zone?
- Figure 1: User - CSP - Scheduling Algorithms
- Problem Description: It is better to represent all the notations and their definitions as a table.
- "cloud providers do not charge for data loads in their infrastructures" - Is it true? If yes, give a proper citation.
- "FL the set of files that must be executed by the available set of virtual machines" - Is the same task carry out in more than one virtual machines?
- Line 184: "The criterion"
- Line 204: "we denoted"
- Example 1: It is similar to the VM placement algorithm in which the VM can be considered as storage only.
- Example 1: The order of assignment from VM1 to VM3 is not clear. Especially, why 7 followed by 3? Why not 3 followed by 7?
- Algorithm 1: Specify the DER(FL) and SCL(FL) process in detail.
- Line 282: ??
- Algorithm 5, Algorithm 6 and Algorithm 7: No need to write both
- Experimental and Discussion: What is Um[.] and UL[.]
- The authors need to compare their algorithm results with any three existing (not yours) algorithms.

Experimental design

- Amazon EC2 is a popular cloud service provider. Why do not the authors consider this case study and implement their solution?

Validity of the findings

no comment

Additional comments

no comment

·

Basic reporting

The paper is overall well written, in some parts there are some short sentences which makes the reading somehow difficult, and I also noticed some repetitions. For example, you state many times that the problem that you try to solve with your algorithms is NP-hard, it is repeated too much times.

For example in the Architecture & Model section, you start again with a literature review while you should start immediately with YOUR model.

In general, the paper is rigourous and clear in formal definitions.

Experimental design

I have a few concerns regarding the problem that you try to solve, it really seems to me that this is a Bin Packing problem that is well-known in literature. Where your problem differs from that?

Then, why your state in the title that your approach is secure? You state a "secure solution"? Where in the algorithms that you propose you guarantee the security? And then security of what? You need to clear identify which kind of security are you dealing with, otherwise remove the term from the title.

When you decide the allocation of these "files"? Then what happens? You wait to collect another set of N files and the start the algorithm over? If a new file arrives and some VM has space why it shouldn't be better to try to allocate it?

Validity of the findings

Novelty of the algorithms should be clearer, in the literature review you should state why the approach that are similar to yours are not good as yours, or why yours is better over the others, why one should use your approach?

Finally I think that a real results discussion is missing, you only report numbers you give no clue about why that numbers are as they are. When you report a result you should explain why you obtained that results, why that approach better or worse, you should expect the results that you obtain otherwise it seems that you runs the algorithms and hope that they work. In tables, put in bold the best value, underline the second and explain why the best is the best and which are the flaws of the others or maybe if they can be good in other contexts or under other circumstances.

---

## Round 0.2 · accepted · Accept

Your manuscript has been accepted for publication in PeerJ Computer Science, according to the comments of the reviewers who evaluated your manuscript. There was one comment that can be addressed before publication.

·

Basic reporting

Good

Experimental design

Good

Validity of the findings

Good

Additional comments

Remove ? from Table 1 (16).

·

Basic reporting

The raised issues have all been addressed correctly.

Experimental design

The raised issues have all been addressed correctly.

Validity of the findings

The raised issues have all been addressed correctly.